# Evaluation of Reference Gene Stability in Goat Skeletal Muscle Satellite Cells during Proliferation and Differentiation Phases

**DOI:** 10.3390/ani14172479

**Published:** 2024-08-26

**Authors:** Siyuan Zhan, Lufei Zhang, Tao Zhong, Linjie Wang, Jiazhong Guo, Jiaxue Cao, Li Li, Hongping Zhang

**Affiliations:** 1Key Laboratory of Livestock and Poultry Multi-Omics, Ministry of Agriculture and Rural Affairs, College of Animal and Technology, Sichuan Agricultural University, Chengdu 611130, China; siyuanzhan@sicau.edu.cn (S.Z.); 18731050023@163.com (L.Z.); zhongtao@sicau.edu.cn (T.Z.); wanglinjie@sicau.edu.cn (L.W.); jiazhong.guo@sicau.edu.cn (J.G.); jiaxuecao@sicau.edu.cn (J.C.); lily@sicau.edu.cn (L.L.); 2Farm Animal Genetic Resources Exploration and Innovation Key Laboratory of Sichuan Province, Sichuan Agricultural University, Chengdu 611130, China

**Keywords:** goat, reference gene, skeletal muscle satellite cell, RT-qPCR

## Abstract

**Simple Summary:**

Investigating the proliferation and differentiation of skeletal muscle satellite cells (MuSCs) in vitro is crucial for elucidating the underlying mechanisms of skeletal muscle development in goats. Real-time quantitative PCR (RT-qPCR) is a widely utilized technique for quantifying the expression levels of target genes. A significant challenge associated with this method is the identification of optimal reference genes for accurate normalization. In this study, we evaluated ten candidate reference genes for the standardization of gene expression. Our results indicated that RPL14 and RPS15A constituted the most stable reference gene combination during the proliferation and differentiation of goat MuSCs.

**Abstract:**

The process of skeletal muscle development is intricate and involves the regulation of a diverse array of genes. Accurate gene expression profiles are crucial for studying muscle development, making it essential to choose the right reference genes for real-time quantitative PCR (RT-qPCR). In the present study, eight candidate reference genes were identified from our previous transcriptome sequencing analysis of caprine skeletal muscle satellite cells (MuSCs), and two traditional reference genes (ACTB and GAPDH) were assessed. The quantitative levels of the candidate reference genes were determined through the RT-qPCR technique, while the stability of their expression was evaluated utilizing the GeNorm, NormFinder, BestKeeper, and RefFinder programs. Furthermore, the chosen reference genes were utilized for the normalization of the gene expression levels of PCNA and Myf5. It was determined that conventional reference genes, including ACTB and GAPDH, were not appropriate for normalizing target gene expression. Conversely, RPL14 and RPS15A, identified through RNA sequencing analysis, exhibited minimal variability and were identified as the optimal reference genes for normalizing gene expression during the proliferation and differentiation of goat MuSCs. Our research offers a validated panel of optimal reference genes for the detection of differentially expressed genes in goat muscle satellite cells using RT-qPCR.

## 1. Introduction

Goat is an economically important animal in many communities worldwide, providing meat and milk products with abundant nutritional value [1]. The intricate process of skeletal muscle growth and development encompasses the activation of skeletal muscle satellite cells (MuSCs), the proliferation and differentiation of myoblasts, and the fusion of myoblasts into myofibers, along with the expression of a diverse array of genes [2,3,4]. The examination of gene expression patterns in goat muscle tissues and cells is crucial for identifying pivotal functional genes that impact meat production traits. This knowledge is imperative for marker-assisted selection (MAS) investigations and for improving breeding initiatives.

The RT-qPCR technique is commonly utilized for assessing gene expression due to its robust specificity, heightened sensitivity, and reliable repeatability [5,6,7]. Nonetheless, the accuracy of RT-qPCR results is heavily reliant on the stability of reference genes [8]. An optimal reference gene is anticipated to exhibit consistent expression levels across diverse experimental conditions, including varying developmental stages, tissue types, and cell types. Nevertheless, the universality of reference gene expression is not absolute, as variations can be observed between different tissues, cell types, and stages of cellular growth [9,10,11,12]. The absence of a universally applicable reference gene necessitates careful consideration in selecting an appropriate reference gene for specific cell and tissue types to avoid potential inaccuracies in results and conclusions [13]. Therefore, the selection of suitable reference genes is crucial for an accurate calibration and normalization of target gene expression under specific experimental conditions. Currently, there is a limited number of systematic studies on reference genes in goat muscle cells, despite various scholars utilizing different algorithms to assess reference genes suitable for livestock muscle tissues [14,15,16,17,18]. Furthermore, the internal regulatory mechanism underlying myogenesis in goats remains poorly understood. Therefore, it is crucial to identify appropriate reference genes in order to investigate the internal regulatory mechanism driving this process further.

In this study, ten reference genes (*HSPA9*, *DDOST*, *RPL5*, *CAPNS1*, *YBX1*, *EEF1G*, *RPS15A*, *RPL14*, *GAPDH*, and *ACTB*) in goat skeletal muscle satellite cells (MuSCs) were examined during the proliferative phase (GM phase) and differentiation phases at day 1 (DM1) and day 5 (DM5). The expression stability of these candidate reference genes was assessed using three established algorithms (GeNorm [19], NormFinder [20], Bestkeeper [21], and RefFinder [22,23]) to gain insights into the proliferation and differentiation processes of goat skeletal muscle satellite cell models in vitro.

## 2. Materials and Methods

### 2.1. Ethics Statement

The Animal Care and Use Committee of the College of Animal Science and Technology, Sichuan Agricultural University, Sichuan, Chengdu, China, approved all of the animal care, slaughter, and experimental procedures in accordance with the Regulations for the Administration of Affairs Concerning Experimental Animals (Ministry of Science and Technology, China) [Approval No. SAU2022302131].

### 2.2. Isolation and Culture of Goat MuSCs

The primary MuSCs were isolated and cultured from *longissimus dorsi* muscle derived from a fetal goat (Chengdu Ma goat, female, *n* = 1), as described previously [24,25]. The cells were cultured in DMEM/F12 (Gibco, Shanghai, China) in a 5% CO_2_ and 95% oxygen incubator at 37 °C, supplemented with 10% FBS (Gibco, Shanghai, China) and 1% penicillin/streptomycin (Solarbio, Beijing, China). When the confluence reached about 80–90%, the MuSCs were seeded into six-well plates. The cells were collected when they achieved about 80–90% confluence in six-well plates in growth medium with three biological replicates, deemed as the proliferation phase (GM). When the cells reached about 80–90% confluence, we replaced the growth medium with differentiation medium containing DMEM/F12, 2% horse serum (Gibco, Shanghai, China), and 1% penicillin/streptomycin. The cells were collected on day 1 (DM1) and day 5 (DM5), with three biological replicates at each time point.

### 2.3. Selection of Candidate Reference Genes

Candidate reference genes were identified from RNA-seq data (PRJNA779184) of goat skeletal muscle satellite cells cultured under proliferation (GM) and differentiation (DM1/DM5) conditions. The selection process was based on the values of the coefficient of variation (CV, %) and fragments per kb per million reads (FPKM). Genes meeting the criteria of FPKM > 100 and CV < 10% were ranked in ascending order based on CV values, with the top eight genes chosen as candidate reference genes. Additionally, two traditional reference genes, GAPDH and ACTB, were also included in the analysis.

### 2.4. RNA Extraction and cDNA Synthesis

Total RNA was extracted from goat MuSCs using the Orizol Chloroform-Free RNA Extraction Kit (Oriscience, Chengdu, China) according to instructions. The integrity and concentration of the RNA were determined by agarose gel electrophoresis (Bio-Rad, Richmond, VA, USA) and a NanoDrop 2000 spectrophotometer (Thermo Scientific, Waltham, MA, USA). Then, the cDNA was synthesized using HiScript IV RT SuperMix for qPCR (+gDNA wiper) (Vazyme, Nanjing, China) following the instructions and stored at −20 °C for backup.

### 2.5. RT-qPCR Analysis

The primers for ten reference genes (*HSPA9*, *DDOST*, *RPL5*, *CAPNS1*, *YBX1*, *EEF1G*, *RPS15A*, *RPL14*, *GAPDH*, and *ACTB*) were designed using the Primer-Blast program (NCBI tools). The sequences of the primer pairs are outlined in Appendix A. Agarose gel electrophoresis and melting curve analysis were conducted to assess the specificity of the primer pairs. The RT-qPCR was performed on the CFX96 Real-Time PCR Detection System (Bio-Rad, Hercules, CA, USA) using 2× M5 HiPer SYBR Permix EsTaq (Mei5bio, Beijing, China). Three biological replicates were performed, and each RT-qPCR was performed in technical triplicates. The reaction conditions included an initial denaturation at 95 °C for 3 min, followed by 39 cycles of denaturation at 95 °C for 30 s and annealing at a specific temperature for 30 s. The standard curve of the RT-qPCR was established using the gradient-diluted cDNA. The correlation coefficient and amplification efficiency were calculated using the CFX Manager Software version 3.1 (Bio-Rad, Hercules, CA, USA). The cycle quantification (Cq) values were automatically generated using the default settings of the Real-Time System.

### 2.6. Stability Analysis of Reference Genes

The expression stability of selected reference genes were evaluated using three programs: GeNorm, NormFinder, and Bestkeeper, following the instructions. The relative expression quantity (Q) of each candidate reference gene was calculated as follows: Q = 2^−ΔCq^, ∆Cq = Cq _(sample)_ − Cq _(minimum)_, where Cq _(sample)_ was the Cq value of a factor in each sample and Cq _(minimum)_ was the minimum Cq value of this gene in all samples [26].

The GeNorm algorithm employs pairwise comparisons to assess each gene’s expression stability, represented by an M value. A lower M value suggests higher stability. GeNorm was also used to determine the minimum number of reference genes required for accurate normalization. NormFinder was used to calculate stability values for candidate reference genes by analyzing their intragroup and intergroup variation. The reference gene with the lowest stability value was considered to be the most stable gene. BestKeeper was used to analyze the expression stability of candidate reference genes by calculating the coefficient of variance (CV) and the standard deviation (SD) based on the raw Cq values. The reference genes with the highest stability had the lowest values of CV and SD, while harboring the highest value of correlation coefficient (r).

### 2.7. Validation of Reference Gene Expression

Target genes were chosen to validate the stability of the reference gene by comparing their expression patterns after normalization. *PCNA* and *Myf5* were selected as target genes because they are the marker genes for MuSC proliferation and differentiation. The expression of target genes was normalized using the most stable and the least stable reference genes. All samples were evaluated in triplicate, and their relative expression levels were calculated using the 2^−ΔΔCt^ method [27].

### 2.8. Statistical Analysis

The results are expressed as means ± standard error of the mean (SEM). All data were evaluated using one-way ANOVA and Duncan’s new multiple range tests using SAS software version 9.2 (SAS, Cary, NC, USA), and *p*-values lower than 0.05 were considered statistically significant.

## 3. Results

### 3.1. Proliferation and Differentiation of Goat MuSCs In Vitro

Quiescent MuSCs were converted to myoblasts and subsequently allowed to proliferate in growth medium (GM) until reaching approximately 80% confluence, marking the proliferation phase as depicted in Figure 1A. Following the replacement of GM with differentiation medium (DM), the cells differentiated into elongated myocytes within one day (Figure 1B). By the fifth day, the myocytes had fused, forming long, multinucleated myotubes (Figure 1C). Cells from the GM, DM1, and DM5 phases were harvested for further experimentation.

### 3.2. RNA Purity, Primer Verification, and Amplification Efficiency

The RNA samples exhibited OD260/280 ratios ranging from 1.99 to 2.06 and RNA integrity number (RIN) values ranging from 8.2 to 9.4 (Appendix A), suggesting high quality and suitability for subsequent experimentation. Analysis of Appendix A revealed successful amplification of a single band of expected size on agarose gel, as well as detection of a single peak in the melting curve for each primer pair, indicating high specificity in amplifying the target fragment across all 10 primer pairs. Furthermore, the standard curves of the reference genes that were tested exhibited strong linear relationships, with amplification efficiency falling within the range of 90.0% to 103.7% and coefficients of determination (R^2^) ranging from 0.982 to 0.999 (Appendix A). These results suggest that the primers performed effectively in the RT-qPCR amplification conditions, producing precise and dependable outcomes.

### 3.3. Analysis of the Expression Levels of the Candidate Reference Genes

The average Cq values of candidate reference genes in all cDNA samples ranged from 15.27 to 20.75 (Figure 2). A lower dispersion of the Cq value indicates higher stability of the gene. Specifically, *RPL14* showed the lowest Cq dispersion (Cq values ranged from 15.27 to 16.34), followed by *RPL5*, *RPS15A*, and *YBX1*, while *ACTB* exhibited the highest variation (Cq values ranged from 15.79 to 18.46). These findings suggest that *RPL14* is the most stable gene in terms of mRNA expression levels, whereas *ACTB* is the least stable.

### 3.4. GeNorm Analysis

The GeNorm program was utilized to calculate the expression stability value (M) of reference genes, with a smaller M value, indicating greater stability. Results revealed that *RPL14* and *RPS15A* exhibited the lowest M values, while *ACTB* had the highest (Figure 3A). This suggests that *RPL14* and *RPS15A* were the most stable internal reference genes, whereas *ACTB* was the least stable. Additionally, we also used GeNorm to analyze the optimal number of reference genes required in this study. GeNorm analysis determined that two reference genes were adequate for accurate normalization of target gene expression, as indicated by V2/3 = 0.03 < 0.15 (Figure 3B).

### 3.5. NormFinder Analysis

The reference genes with the lowest stability values, as determined by the NormFinder method, were deemed to be the most stable. The stability of ten candidate reference genes were then ranked based on the results of the NormFinder analysis (Figure 4). The two most stable reference genes screened by the software were *RPL14* (0.026) and *HSPA9* (0.037), and *ACTB* (0.560) was the least stable reference gene.

### 3.6. Bestkeeper Analysis

Bestkeeper software (version 1) estimates the stability of reference genes based on the coefficient of variation (CV) and standard deviation (SD). A lower CV or SD value indicates a more stable reference gene. In this study, the most stable reference genes (*RPS15A* and *YBX1*) and the least stable gene (*ACTB*) were identified based on stability rankings (Table 1).

### 3.7. Comprehensive Analysis of Candidate Reference Genes

To combine the results of analysis by the three programs, the stability ranks of the candidate reference genes were calculated and ranked by geometric mean (Table 2) and RefFinder program (Figure 5). According to the comprehensive stability rankings, *RPL14* and *RPS15A* were the two most stable reference genes, and *ACTB* was the least stable reference gene. Therefore, *RPL14* and *RPS15A* were the most appropriate reference genes to normalize the expression of target genes during the proliferation and differentiation phases of goat MuSCs in vitro.

### 3.8. Expression Validation of Candidate Reference Genes Using Target Genes

In order to further validate the selection of candidate reference genes, the most stable reference genes (*RPL14* and *RPS15A*) as well as the least stable reference gene (*ACTB*) were utilized to normalize the same target genes. The mRNA expression of *PCNA* in goat MuSC during the proliferation phase (GM) and differentiation phases (DM1 and DM5) showed a significant difference (*p* < 0.01) when normalized using *RPL14* and *RPS15A* either individually or in combination. However, there was no significant difference observed between GM and DM5 when normalized using *ACTB* (Figure 6A). During the transition from the proliferation phase to the differentiation phases of goat muscle satellite cells (MuSCs), the mRNA expression of *Myf5* exhibited an initial increase followed by a decrease when normalized using *RPL14* and *RPS15A* as reference genes, either individually or in combination (Figure 6B). Conversely, normalization using *ACTB* as the reference gene resulted in an anomalous increase in *Myf5* mRNA expression between the GM phase and the DM1 phase (Figure 6B). This discrepancy underscores the potential for misinterpretation of target gene expression when inappropriate reference genes are employed. Consequently, the selection of stable and appropriate reference genes is crucial for the accurate normalization of relative gene expression levels.

## 4. Discussion

The analysis of gene expression is a common measurement in molecular studies, and the current gold standard protocol is RT-qPCR [28,29,30,31,32]. This method requires an accurate and reliable reference gene to standardize the relative expression level of specific target genes [33]. In this regard, the significance of reference genes for the accurate analysis of target gene expression is well established. Ideal reference genes are expected to exhibit stable expression levels across all experimental conditions as well as in various tissues or cell types. However, it has been clearly demonstrated that there is no universal reference gene that maintains stable expression under all conditions [34,35]. In most studies of MuSCs, reference genes are typically selected based on literature reviews and prior experience with other organisms or tissues rather than on empirical evidence supporting their efficacy. Consequently, it is generally recommended that reference genes be validated for each species and specific experimental conditions. This validation is essential for the accurate measurement of gene expression using RT-qPCR, as the selection of inappropriate reference genes may lead to inaccurate values or even contradictory results [36,37,38].

Transcriptome sequencing is a vital research method for analyzing gene expression and is employed to identify differentially expressed and functional genes. Notably, utilizing transcriptome data to screen for reference genes is an effective experimental approach for selecting reference genes in non-model species [39]. In this study, eight candidate reference genes were identified based on our previous RNA-seq data of goat skeletal muscle satellite cells (MuSCs). Additionally, two traditional reference genes were assessed. The Cq values of the candidate reference genes were determined using RT-qPCR, and the stability of their expression was evaluated utilizing the GeNorm, NormFinder, BestKeeper, and RefFinder programs. Our study comprehensively identified *RPL14* and *RPS15A* as the two most stable reference genes during the proliferation and differentiation of goat MuSCs. These genes exhibited greater stability compared to *GAPDH* and *ACTB*, indicating that traditional reference genes may not be suitable for certain situations. Previous studies have demonstrated that *GAPDH* and *ACTB* are not suitable for the normalization of skeletal muscle development in cattle, with analogous findings reported in pigs, goats, and mice [18,40,41,42], corroborating the results of the present study. Furthermore, our comprehensive ranking analysis indicates that *GAPDH* exhibits greater stability than *ACTB* during in vitro differentiation of goat muscle satellite cells. *RPL14* and *RPS15A* are members of the ribosomal protein family. Ribosomal proteins play crucial housekeeping roles in ribosomal biogenesis and protein synthesis and are essential for cell growth, proliferation, differentiation, and development [43,44]. *RPL32* and *RPS18* have been identified as suitable reference genes for the development of skeletal muscle in pigs [18]. Additionally, a study suggested that *RPS4X* and *RPS6* are stably expressed during rumen development in goats [45]. *RPS15A* has been determined to be the most appropriate reference gene during the proliferation and differentiation phases of bovine skeletal muscle satellite cells in vitro [46]. Previous studies have demonstrated that *RPLP0* is one of the reference genes proposed for data normalization in bovine muscle [47]. *RPL15* has been identified as the most stable reference gene in bovine oocytes collected during both winter and summer [48], whereas *RPL4* has shown the highest stability as a reference gene during the differentiation of bovine bone marrow mesenchymal stem cells [49]. Additionally, *RL13A* has been consistently expressed as a stable reference gene for gene expression normalization across various muscle tissues in domestic yaks [50]. Consistent with the results of the present study, these findings suggest that ribosomal protein family genes have the potential to serve as stable internal reference genes. However, it has also been demonstrated that, despite their stability, ribosomal protein family genes may not be universally reliable as reference genes [51]. Therefore, it is essential to evaluate the expression stability of these genes under specific experimental conditions.

To our knowledge, this study represents the first validation of the expression stability of reference genes during the proliferation and differentiation phases of goat muscle satellite cells in vitro. Furthermore, we determined the optimal combination of stably expressed reference genes and identified the least stable ones for use during the in vitro proliferation and differentiation induction of goat MuSCs. Our findings provide a critical reference for selecting appropriate reference genes for gene expression analysis via RT-qPCR in future studies involving goat MuSCs.

## 5. Conclusions

In summary, *RPL14* and *RPS15A* were identified as the most suitable reference genes for RT-qPCR experiments in goat MuSCs during proliferation and differentiation in vitro. Furthermore, utilizing a combination of *RPL14* and *RPS15A* is recommended as the optimal method for normalizing the expression levels of target genes in goat MuSC experiments.

## Figures and Tables

**Figure 1 animals-14-02479-f001:**
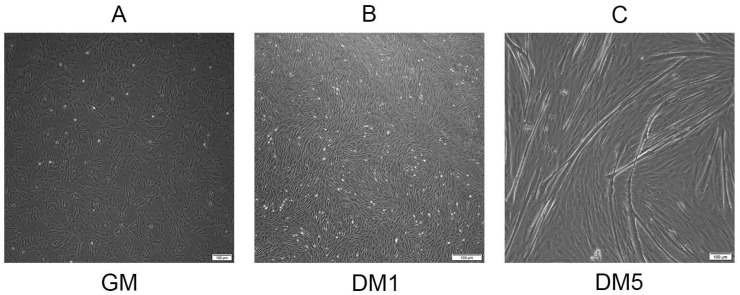
Proliferating and differentiating goat skeletal muscle satellite cells in vitro. (**A**) MuSCs were cultured in the growth medium until they achieved 80% confluence (GM). (**B**) MuSCs were cultured in the differentiation medium for 1 day (DM1). (**C**) MuSCs were cultured in the differentiation medium for 5 days (DM5). Scale bars = 100 µm.

**Figure 2 animals-14-02479-f002:**
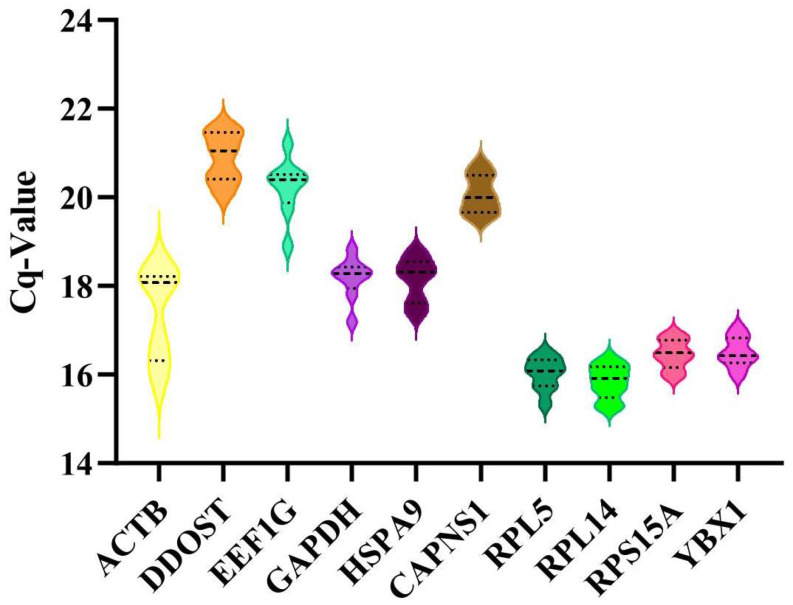
Violin plot of Cq values of ten candidate reference genes obtained from all cDNA sample.

**Figure 3 animals-14-02479-f003:**
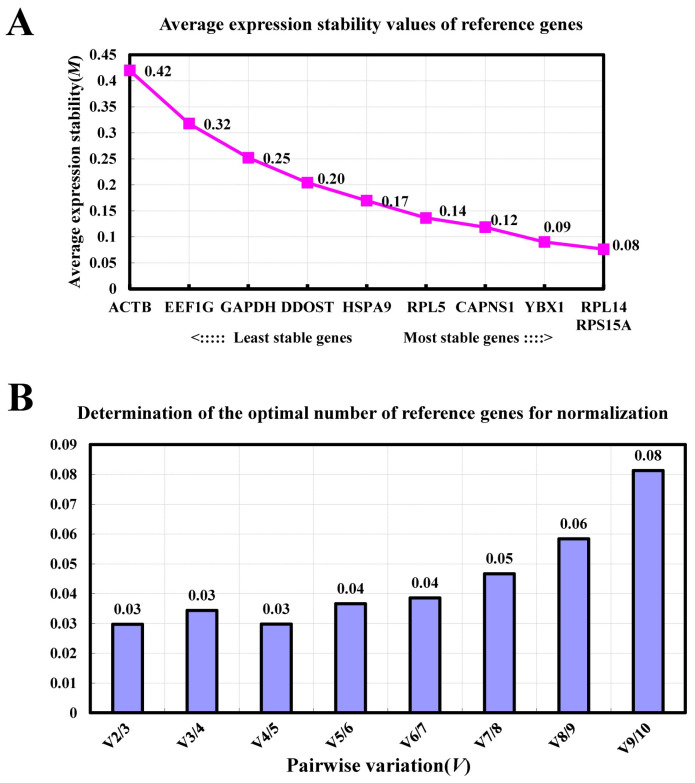
Expression stability and the optimal number of candidate reference genes were determined using the GeNorm program. (**A**) Stability of candidate reference genes during proliferation and differentiation phases of goat MuSCs in vitro. (**B**) Determination of the optimal number of reference genes required for normalization.

**Figure 4 animals-14-02479-f004:**
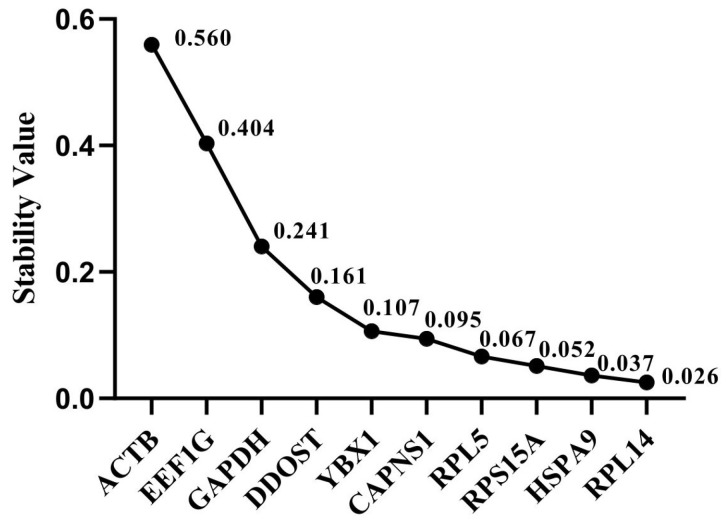
Expression stability of candidate reference genes as analyzed using the NormFinder program.

**Figure 5 animals-14-02479-f005:**
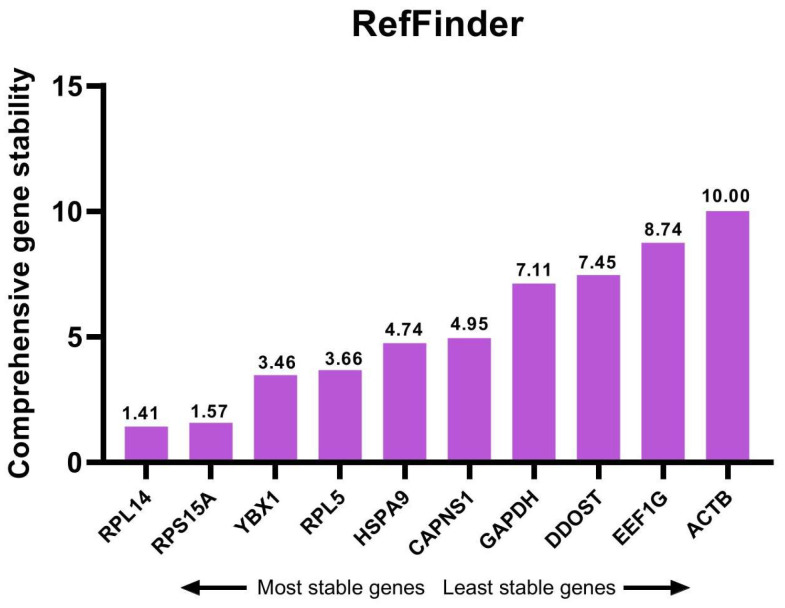
Comprehensive ranking values of candidate reference genes based on RefFinder program.

**Figure 6 animals-14-02479-f006:**
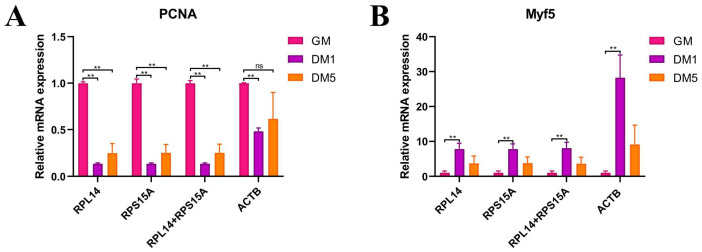
Relative expression of *PCNA* (**A**) and *Myf5* (**B**) normalized to the most stable reference genes (*RPL14*, *RPS15A*, and *RPL14*+*RPS15A* combination) and the least stable gene (*ACTB*) in goat MuSCs. ** *p* < 0.01. “ns” means no significant difference between the two groups.

**Table 1 animals-14-02479-t001:** Expression stability of candidate reference genes estimated by Bestkeeper algorithm.

Gene	Coefficient of Variation (CV)	Standard Deviation (SD)	Rank
*RPS15A*	1.58	0.26	1
*YBX1*	1.66	0.27	2
*RPL5*	1.76	0.28	3
*RPL14*	1.98	0.31	4
*GAPDH*	1.79	0.33	5
*CAPNS1*	1.82	0.36	6
*EEF1G*	2.22	0.45	7
*HSPA9*	2.60	0.47	8
*DDOST*	2.30	0.48	9
*ACTB*	5.39	0.94	10

**Table 2 animals-14-02479-t002:** Stability ranking of candidate reference genes analyzed by the three combined algorithms.

Gene	Program			Mean Rank	Comprehensive Rank
	GeNorm	NormFinder	Bestkeeper		
*RPL14*	1	1	4	1.59	1
*RPS15A*	2	3	1	1.82	2
*YBX1*	3	6	2	3.30	3
*RPL5*	5	4	3	3.91	4
*HSPA9*	6	2	8	4.58	5
*CAPNS1*	4	5	6	4.93	6
*GAPDH*	8	8	5	6.84	7
*DDOST*	7	7	9	7.61	8
*EEF1G*	9	9	7	8.28	9
*ACTB*	10	10	10	10.00	10

## Data Availability

The original contributions presented in the study are included in the article/Appendix A. Further inquiries can be directed to the corresponding author.

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
