# Peer review of "Evaluation of Reference Gene Stability in Goat Skeletal Muscle Satellite Cells during Proliferation and Differentiation Phases"

_animals, 2024, doi:10.3390/ani14172479_

Round 1

Reviewer 1 Report

Comments and Suggestions for Authors

The authors evaluated the reference gene stability in goat skeletal muscle satellite cells and found that RPL14 and RPS15A are the most stable reference gene combination of goat muscle satellite cells.
They analyze eight candidate reference genes obtained from RNA-seq data and two traditional reference genes ACTB and GAPDH. In qRT-PCR experiments, it is known that some reference genes may not have constant expression levels in a tissue- or cell-specific manner, so using these genes for normalization may affect the original gene expression analysis. The use of appropriate reference genes is therefore very important to ensure the accuracy of the experiment. In this manuscript, traditional and very widely used reference genes, ACTB and GAPDH, are not suitable for normalizing target gene expression in goat skeletal muscle satellite cells proliferation and differentiation. These results will provide useful information for future gene expression experiments using goat muscle satellite cells.

The manuscript is well written and the selection criteria for the ten candidate reference genes also appear to be acceptable. The results were enough to support the conclusions.

Only one point to be made.
-Supplementary Figure S1
In the GADPH gel photo, the lane is cut and pasted, so it is not suitable. Please use the appropriate photo like other genes.

Author Response

The manuscript is well written and the selection criteria for the ten candidate reference genes also appear to be acceptable. The results were enough to support the conclusions.

Response: Thanks for your positive comments.

Comments 1: Only one point to be made.Supplementary Figure S1. In the GADPH gel photo, the lane is cut and pasted, so it is not suitable. Please use the appropriate photo like other genes.

Response 1: Thank you for pointing out this issue, it has been corrected in the revised manuscript. We have reattached the complete and original GAPDH gel photo in Supplementary Figure S1.

Reviewer 2 Report

Comments and Suggestions for Authors

I have only minor comments on this well-written manuscript.  You conclusions are justified by the data and discussion presented.  Here are a couple items.

line 77:  Indicate that the other 95% was oxygen. 

Figure 2 :  Can you enlarge the labels on the axes for both A and B so that they are readable.  

Discussion:  Would have been helpful if you had made a comparison of the stability of GAPDH and perhaps actin as well.  Any comment?

References:  Make sure that capitalizations of titles of references are consistent.

Author Response

I have only minor comments on this well-written manuscript. Your conclusions are justified by the data and discussion presented. Here are a couple items.

Response: Thanks for your positive comments.

Comments 1: line 77:  Indicate that the other 95% was oxygen.

Response 1: Thanks for your suggestions. It has been added in the revised manuscript (line 88).

Comments 2: Figure 2 : Can you enlarge the labels on the axes for both A and B so that they are readable. 

Response 2: Thanks for your suggestions. Figure has been updated in the revised manuscript.

Comments 3: Discussion: Would have been helpful if you had made a comparison of the stability of GAPDH and perhaps actin as well. Any comment?

Response 3: Thanks for your suggestions. We have made modifications based on your comments, please see the discussion section for details.

Comments 4: References: Make sure that capitalizations of titles of references are consistent.

Response 4: Thanks for your suggestions. We have checked the format of the references according to your comment, and confirmed that capitalizations of titles of references are consistent.

Reviewer 3 Report

Comments and Suggestions for Authors

Dear authors,

I will not recommend your manuscript for publication, because you results not be significant and verified. 

L58 – GAPDH and ACTB not candidate genes. These were appropriated reference genes. Correct it.

L75 – Are you using only one origin of MuSCs? It is wrong. If you are wanting to detect new reference genes for genes expression evaluation, then representative selection must include some cell lines from different organisms. You result is significant only for you research and need to be tested in next research. According you results many research worldwide must be named “unsignificant”, because GAPDH and ACTB genes were used as reference. Please, be careful with your conclusions.

Not clear – how many samples were used for PCR? I understand – 3. One in GM, one in DM1 and one in DM5. It is not good for reference genes search.

L273 – I was not found in Discussion information of some research, where GAPDH and ACTB genes were not be considering as reference genes. May be some genes were considered as stable, but it is not recommended to no using GAPDH and ACTB in research.

L298 – According to investigation of only one organism MuSCs, you are can declare only two new candidate reference genes, not more. But it is declaration must be so carefully – to small significance of this research.

Regards,

Author Response

Comments 1: L58 – GAPDH and ACTB not candidate genes. These were appropriated reference genes. Correct it.

Response 1: Thanks for your comments, it has been corrected in the revised manuscript.

Comments 2: L75 – Are you using only one origin of MuSCs? It is wrong. If you are wanting to detect new reference genes for genes expression evaluation, then representative selection must include some cell lines from different organisms. You result is significant only for you research and need to be tested in next research. According you results many research worldwide must be named “unsignificant”, because GAPDH and ACTB genes were used as reference. Please, be careful with your conclusions.

Response 2: Thanks for your comments. As the optimal reference genes for different species and different tissues are not necessarily the same, it is essential to determine the optimal reference genes for specific species used in the experimental system. The purpose of this study is to identify stable and appropriate internal reference genes for subsequent goat MuSCs research. Therefore, we used goat-derived skeletal muscle satellite cells. Frequently, GAPDH and ACTB are selected as reference genes without proper validation. Such unvalidated choices can compromise the precision of normalization and cast doubt on the accuracy of gene expression estimates. This underscores the critical need to validate the selected reference genes for each study. GAPDH and ACTB, as traditional internal reference genes, do not always show good expression stability in different species, tissues and cells. In this study, we found that GAPDH and ACTB was not suitable for RT-qPCR experiments in goat MuSCs during proliferation and differentiation in vitro. Therefore, our conclusions are only applicable to relevant studies focusing on goat skeletal muscle satellite cells.

Comments 3: Not clear – how many samples were used for PCR? I understand – 3. One in GM, one in DM1 and one in DM5. It is not good for reference genes search.

Response 3: Thanks for your comments. In the seciton 2.2, we indicate the sample size. Line 92-93, The cells were collected when they achieved about 80-90% confluence in six-well plates in growth medium with three biological replicates, deemed as proliferation phase (GM). Line 96-97, The cells were collected at day 1 (DM1) and day 5 (DM5), with three biological replicates at each time point.

In addition, in section 2.5, we added this sentence based on your comment. “Three biological replicates were performed and each RT-q PCR was performed with tech-nical triplicates.”(Line 120-121).

Comments 4: L273 – I was not found in Discussion information of some research, where GAPDH and ACTB genes were not be considering as reference genes. May be some genes were considered as stable, but it is not recommended to no using GAPDH and ACTB in research.

Response 4: Thanks. The Discussion section has been modified appropriately based on your comments. GAPDH and ACTB, as traditional internal reference genes, do not always show good expression stability in different species, tissues and cells. In this study, we found that GAPDH and ACTB was not suitable for RT-qPCR experiments in goat MuSCs during proliferation and differentiation in vitro. Previous studies have demonstrated that GAPDH and ACTB are not suitable for the normalization of skeletal muscle development in cattle, with analogous findings reported in pigs, goats, and mice [1-4], corroborating the results of the present study.

  1. Niu, G.L.; Yang, Y.L.; Zhang, Y.Y.; Hua, C.J.; Wang, Z.S.; Tang, Z.L.; Li, K. Identifying suitable reference genes for gene expression analysis in developing skeletal muscle in pigs. Peerj 2016, 4, e2428.
  2. Niemann, H.; Najafpanah, M.J.; Sadeghi, M.; Bakhtiarizadeh, M.R. Reference Genes Selection for Quantitative Real-Time PCR Using RankAggreg Method in Different Tissues of Capra hircus. PLoS ONE 2013, 8.
  3. Saremi, B.; Sauerwein, H.; Danicke, S.; Mielenz, M. Technical note: identification of reference genes for gene expression studies in different bovine tissues focusing on different fat depots. J Dairy Sci 2012, 95, 3131-3138.
  4. Thomas, K.C.; Zheng, X.F.; Garces Suarez, F.; Raftery, J.M.; Quinlan, K.G.; Yang, N.; North, K.N.; Houweling, P.J. Evidence based selection of commonly used RT-qPCR reference genes for the analysis of mouse skeletal muscle. PLoS One 2014, 9, e88653.

Comments 5: L298 – According to investigation of only one organism MuSCs, you are can declare only two new candidate reference genes, not more. But it is declaration must be so carefully – to small significance of this research.

Response 5: Thanks for your comments. In this study, RPL14 and RPS15A were identified as the most suitable reference genes for RT-qPCR experiments in goat MuSCs during proliferation and differentiation in vitro. Furthermore, utilizing a combination of RPL14 and RPS15A is recommended as the optimal method for normalizing the expression levels of target genes in goat MuSC experiments. Our findings provide a critical reference for selecting appropriate reference genes for gene expression analysis via RT-qPCR in future studies involving goat MuSCs.

Round 2

Reviewer 3 Report

Comments and Suggestions for Authors

Dear authors,

I still not appropriate significance of your research, but if respecting Editor wwas send me this manuscript for second round review, then I confirm his mention.

Regards,